# OpenReview forum: "Measuring AI Ability to Complete Long Software Tasks"
_NeurIPS.cc/2025/Conference — NeurIPS 2025 poster_

### Official Review · Reviewer_npLT · 2025-06-26

**Clarity:** 4
**Significance:** 4
**Originality:** 3
**Rating:** 6
**Confidence:** 4

**Summary:**

The paper evaluates the performance on agentic tasks (including e.g. SWE and ML tasks) of 12 models released over the past 6 years, and analyses how models get better over time. To do that, it uses RE-Bench, HCAST and a newly created set of short tasks called Software Atomic Actions, and draws on extensive human baseline data comprising hundreds of human baseline trajectories taking in total >2,500 hours by skilled SWEs and ML researchers. Based on this data, each task is assigned a task length (how long humans took on average to solve it). The paper’s main contributions are: a) The definition of a model’s time horizon as the (estimated) length of tasks which the model can do with 50% reliability. This is an intuitive measure of a model’s agentic capabilities. b) The discovery of an exponential trend in models’ time horizons: since 2019, models’ time horizons have doubled approximately every 7 months. The trend approximately holds up under various ablations and can be used to forecast the arrival of future highly agentic models.

**Questions:**

- IIUC, the human baseline methodology is slightly different between the different datasets (SWAA, HCAST, RE-Bench). How worried are the authors that these differences systematically bias the fitted trend?
- How far do the authors think the trend will continue? Does it seem possible that e.g. week-long or month-long tasks will tend to be more messy, and therefore the trend fitted on non-messy tasks won’t hold up?
- (Nit) Table 2 would be more informative if the analysis was done on the same set of tasks

**Ethical Concerns:**

["NO or VERY MINOR ethics concerns only"]

**Final Justification:**

Very strong paper and thorough rebuttal.

**Limitations:**

yes

**Paper Formatting Concerns:**

no major formatting issues

**Quality:**

4

**Strengths And Weaknesses:**

Strengths
- This is a highly significant and original model evaluation paper. Most evaluation papers focus on a narrow capability and are quickly saturated, as a result of which they cannot be used for tracking and forecasting model capabilities across multiple years and model generations. The discovered trend is simple to understand and highly consequential for timelines to AGI and beyond, and any efforts aimed at preparing society for that transition.
- The human benchmarking effort that went into this paper (as well as HCAST and RE-Bench), is immense, especially in light of the fact that all the baseliners were experienced SWEs / ML researchers / cybersecurity experts. This makes the time-horizon metric grounded and highly interpretable.
- The paper is very rigorously done. It has many ablations, e.g. accounting for task messiness, a sensitivity analysis for the final forecast, a follow-up experiment on real-life pull-requests, … Many limitations or follow-up questions I had while reading the paper were discussed in detail in the appendix.
- The writing is clear, high information-density, and the figures / tables are informative.

Weaknesses (minor)
- While the logarithmic axis makes sense for longer time horizons, I’m not sure that the distinction between a 1s, 2s, 4s, 8s, 15s, or 30s task is that meaningful or reliably measurable, and so I’m not sure how seriously to take the results for the early models.
- One interesting observation is that models’ competence at different levels of reliability spans quite a few orders of magnitude in terms of time horizons. E.g. tasks the model can do 90% of the time can be ~60 times shorter than tasks the model can do 10% of the time. An analysis of this in more detail could improve the paper.

---

> ### Author Rebuttal · Authors · 2025-07-31
>
> Thank you for your detailed review of our work! We appreciate that you found our work highly significant and original, and that you found our writing clear and methodology rigorous.
>
> - **Distinction between very short tasks:** We agree that short tasks are difficult to measure, and we attempted to mitigate this by designing the SWAA baselining app for minimal friction. The median SWAA task has a stdev of a factor of 1.43 in its human baseliner time (i.e. median_{t \in tasks} exp(stdev_{all baseline times on task t}(log(human time))) = 1.43), much lower than the range of SWAA task lengths, which is about a factor of 20. Note a limitation of SWAA is that many short segments of intellectual work are not capturable in standalone tasks, so SWAA is still far from a representative distribution of 2-second or 15-second tasks.
> - **Differences between time horizons of different reliability:** We agree that it’s interesting that the tasks model can complete very reliably are so much shorter than tasks they can complete 50% or 20% of the time, though we’ll note that this seems consistent with anecdotal evidence about how current frontier models can be competent on average but are not reliably so. Interestingly, we do observe that 80% horizons are improving at similar rates to 50% horizons (Figure 17), and we find this exponential increase more convincing than the exact time horizon figures.  We’ll edit the paper to include more discussion about these observations.
>
> **Questions:**
>
> - **Differences between SWAA and other baseliners:** The baseline methodologies generally match the ways that humans interface with tasks in the real world but are not perfect. For SWAA, humans immediately type an answer, much like a short segment of SWE work. On HCAST and RE-Bench, the baseliner instructions are somewhat different-- "time to reach target score" vs "maximize score after 8 hours"-- so we wouldn't be surprised if the effective times differed by 25% or so. This error is swamped by variance derived from the limited number of long tasks, but a consistent methodology would be important for more precise time horizon estimates. We also do robustness checks, including fitting a trendline without SWAA (Figure 10) which is consistent with the main trend. In the camera ready, we’ll also have additional SWE-bench Verified data and analysis of GPQA and LiveCodeBench, which use different human baseline or time estimate methodologies.
> We also note that when forecasting, the trendline slope only weakly depends on the bias of human baseline data since the y axis spans orders of magnitude. A 2x bias in SWAA tasks will only affect the projected date of 1-month software AI by a few months.
>
> - **Speculation on future trend:** Predicting the future is always difficult, but we can speculate on the four factors in section E.2. The effectiveness of agency training could push the trend in either direction at any time, but the prevalence of reward hacking behavior suggests it might cause faster progress on verifiable tasks than nonverifiable ones. Compute scaling will probably continue for between 2-5 more years before it slows down the trend. AI R&D automation will probably depend on the ML time horizon reaching some unknown threshold. As for AGI generalizing better than current AI systems, we still think this could happen but have little idea when. Our overall guess is that the future trend on verifiable tasks will be around every 3-5 months, and on realistic tasks about every 4-8 months, with large uncertainties. This will continue until either compute limitations, AI R&D automation, or AGI happen.
>
>   As for the messiness of week+ long tasks, we think the low messiness of our dataset is an important limitation at all task durations. Most week-long tasks will have at least some messy segments, and the time horizon may be much lower on those subtasks leading to higher overall failure rates. It is rare for a software engineer to go a week without encountering poor specifications or resource limits. In some analysis which will be in the camera ready, OSWorld and WebArena (agentic computer use) time horizons are as low as 2-4 minutes, but tooling has improved faster than we thought with systems like ChatGPT Agent, so some types of messiness may have less of an impact.

---

> > ### Comment · Reviewer_npLT · 2025-08-08
> > **Thank you for the detailed rebuttal; I stand by strong accept**
> >
> > Thank you for your detailed answers, and for updating the paper to include the time horizon analysis at different levels of reliability!
> >
> > Short tasks: Thank you for illuminating some details about your benchmarking app, and for extracting statistics. I have updated somewhat towards this being less of a problem than I thought (or at least that you've taken reasonable precautions against this).
> >
> > Speculation on future trend: I'd be curious where the 2-5 years estimate for compute scaling limits comes from? (but no worries if you don't see this in time)
> >
> > Overall I found the authors answers very informative and I'm happy to defend my original assessment (strong accept).

---

### Official Review · Reviewer_woeK · 2025-06-30

**Clarity:** 3
**Significance:** 3
**Originality:** 3
**Rating:** 5
**Confidence:** 4

**Summary:**

This paper introduces a novel measurement for assessing AI capabilities: the 50% task-completion time horizon, i.e. the amount of time it takes a human to complete tasks that the AI model achieves 50% success rate. The authors tested over 100 tasks across three datasets. Based on the results, they conducted backtesting and provided forward-looking analyses, such as estimating when AI systems might be able to complete one-month-long software tasks.

**Questions:**

Considering the high cost of collecting human baseline data, do you have any insights on reducing variance or bias in the measurement process to obtain a more robust result.

**Ethical Concerns:**

["NO or VERY MINOR ethics concerns only"]

**Final Justification:**

I have no further concerns after reading the author's rebuttal, and decide to maintain my positive score of the paper.

**Limitations:**

yes

**Paper Formatting Concerns:**

No paper formatting concerns

**Quality:**

3

**Strengths And Weaknesses:**

Strengths:

1. The proposed 50%-task-completion time horizon is an interesting and reasonable metric for evaluating AI model capabilities. Unlike traditional success rate or performance-based metrics, it better captures model behavior in more realistic and automated settings.

2. The authors validate this metric across 170 diverse tasks and 12 frontier models. The results are reasonable and informative.

3. The authors further apply the metric to perform verification and exploration, including replicating their measurement on SWE-bench Verified and extrapolating toward one-month-horizon AI. These analysis contributes to both a robust measurement approach and valuable insights.

Overall, using time horizons as a metric provides a fresh perspective on evaluating large language models. Although similar ideas were mentioned in the GPT-4o technical report, this paper stands out by offering a more systematic and comprehensive treatment of the concept.

Weaknesses:

1. The human baseline sample size is relatively small, which could bring significant bias into the overall evaluation.

2. The paper may be better suited for a datasets and benchmarks track.

3. The paper lacks discussion of recent works that also explores LLM capability evaluation beyond traditional success-based performance metrics. For example: Miserendino, Samuel, et al. "SWE-Lancer: Can Frontier LLMs Earn $1 Million from Real-World Freelance Software Engineering?." arXiv preprint arXiv:2502.12115 (2025).

---

> ### Author Rebuttal · Authors · 2025-07-31
>
> Thanks for the detailed review of our work! We’re glad that you think our work provides a fresh perspective on evaluating LLMs and that you find our results informative and insightful.
>
> - **Human baseline sample size/population variance:**  We agree that the task lengths measured on different human populations will differ substantially, and we do some robustness checks related to this in Section 4. On SWE-bench Verified (Section 4.1), whose task lengths are estimates of high-context maintainer time, models have shorter time horizons. In our main analysis, the population of human baseliners is people from top-100 universities or equivalent aptitude, with several years of experience in their field (Section 2.2); this is consistent between HCAST, RE-Bench, and SWAA, with details of each in C.1.
> - **SWE-Lancer:** Thanks for bringing up the SWE-Lancer paper! We’ll cite and discuss it in our work.
> - **Reducing variance or bias in measurement data:** We have several ideas for how future work might increase measurement precision:
>   - As we mention in Section 3.2, errors between time horizons are correlated, so relative comparisons have smaller error bars than absolute ones, e.g. a bootstrap test shows o3 is significantly above Claude 3.7 and above trend. We are also somewhat conservative with our CIs as they do 3-level hierarchical bootstrap, which some literature suggests overestimates variance.
>   - Correcting for developer-specific effects (some humans are faster than others): not included because we had too few tasks with multiple human baselines to calibrate them accurately. The baseliner pool has different specialties for different tasks, so the tasks done by different baseliners were often disjoint.
>   - Tasks could have subtasks like Cybench-- not done in this paper because the long tasks were taken from Rein et al. and Wijk et al., and the shorter SWAA tasks are only one step.
>   - Tasks could be stratified by difficulty. This would require a new benchmark constructed to have task variants that are easier vs harder for models at the same length for humans, which would be difficult to keep realistic.
>   - Item response theory methods could be applied to place more importance on more informative tasks. We rejected this for simplicity but it could be worthwhile in other contexts.
>   - Overall we think there isn't enough statistical juice to be worth the complexity; as for future work we think it's best to collect human baseline data from real software development in the wild, which will have much lower marginal cost.

---

### Official Review · Reviewer_irnC · 2025-07-01

**Clarity:** 2
**Significance:** 2
**Originality:** 3
**Rating:** 4
**Confidence:** 3

**Summary:**

- This paper proposes a new metric to evaluate AI abilities, the 50% task-completion time horizon, which is the time humans typically take to complete tasks that AI models can complete with a 50% success rate.
- The authors evaluate a dozen or so models on three datasets and compute the task-completion time horizon metric.
- The authors find that the task-completion time horizon has been growing exponentially in the past few years.

**Questions:**

- Can the authors discuss why this particular metric was selected and how to mitigate concerns about the population of human baseliners that are used?
- Can the authors discuss how benchmarking more agentic frameworks might affect their results?
- Can the authors explain what the primary takeaways from this work are beyond the exponential trends observed in existing data?

**Ethical Concerns:**

["NO or VERY MINOR ethics concerns only"]

**Final Justification:**

Per my review, there are multiple presentation aspects that can be improved in the next version.

**Limitations:**

Yes

**Quality:**

2

**Strengths And Weaknesses:**

Quality:
- I agree with the author's reasoning about why existing metrics are flawed. However, they fail to provide a sufficient justification for why X% task completion time horizon is the “right” metric that the community should be thinking about.
- The reliance of the metric on human baseliners means the results might depend a lot on the population of humans that is recruited. Can the authors comment on this?

Clarity:
- Since the authors focus on software tasks, why do you only evaluate LLMs on their own? Many software tasks (e.g., SWE-Bench) cannot be solved with one LLM call and rather need more agentic frameworks.
- While the authors claim to focus on realistic tasks, the collection seems like a somewhat adhoc combination of software-related tasks. More discussion on how the set of tasks covers software engineering development and life cycle would be helpful.
- I noticed there is consistent writing. For example, L47 states that 13 models were evaluated, L111 says 12 frontier and 3 “near frontier” models, and then L256 states 11 models. The inconsistencies throughout this paper are concerning.

Significance:
- I am finding it difficult to understand what the significant takeaways or contributions are of this work and how it implicates future work. The primary contribution of the work seems to be evaluating models on existing datasets and baseliner results with relatively simple methods.

Originality:
- This type of metric seems fairly novel in the literature. Most other evaluations do not have an axis of human task completion time.

---

> ### Author Rebuttal · Authors · 2025-07-31
>
> Thanks for your review!
>
> We’d like to start by clarifying that our work is studying the ability of AI agents to perform agentic tasks – both RE-Bench and HCAST tasks require models to take multiple actions in series, and many of the agent runs use millions of tokens. We discuss these tasks in more detail in Section 3.1 and the ReAct-based agent scaffolds we used in Sections 3.3 and C.3. We agree that we could’ve been more clear that we’re studying AI performance on agentic tasks in the introduction and abstract; we’ll revise the terminology there to be more clear and consistent with the rest of the paper.
>
> We also apologize for the typos on the number of models. In total, we evaluated 16 models (Figure 25), of which 12 advanced the state-of-the-art in terms of time horizon and 4 did not (and thus were ‘near-frontier’). We’ll edit the text to clarify this; thanks for pointing out these errors.
>
> To address some other questions throughout the review:
> 1. **Population of human baseliners:** We agree that the task lengths measured on different human populations will differ substantially, and we do some robustness checks related to this in Section 4. On SWE-bench Verified (Section 4.1), whose task lengths are estimates of high-context maintainer time, models have shorter time horizons. We also measure the ratio of time taken by contractors vs maintainers on our internal PRs (Section C.2). In our main analysis, the population of human baseliners is people from top-100 universities or equivalent aptitude, with several years of experience in their field (Section 2.2); this is consistent between HCAST, RE-Bench, and SWAA, with details of each in C.1.
> 2. **Task distribution vis-a-vis SWE development cycle:** The HCAST (Rein et al) and RE-Bench (Wijk et al) papers have a breakdown of the categories each task falls under. HCAST is 30% MLE, 20% SWE, 20% cybersecurity, and 30% general CS-relevant reasoning, with subcategories given in Table 1 of the HCAST paper. RE-Bench is 7 long ML research engineering settings, each of which inherently require subskills like algorithmic thinking and debugging. As for SWAA, a breakdown between math, code completion, and other decisions is in appendix B.1.3. Overall, we have coverage of several SWE subfields and process steps, but lack the ability to measure some elements of such work that are inherently difficult to benchmark, like project management and code quality.
> 3. **Choice of time horizon metric:** We chose the time horizon metric due to its relevance to the role of AI in the future economy. When companies choose whether to automate a task, two important factors are the amount of human labor the AI can replace with minimal supervision, and the reliability of the AI. The time horizon metric combines these into a measure of performance that should relate to which tasks must still be done by human engineers vs. can be automated. Time horizon also has good explanatory power for the reliability of agents at different task lengths (Figure 4); the logistic fit is reasonably good and we find that agents’ 80% time horizons are roughly 5x shorter than their 50% horizons (Figure 17).
> 4. **Other takeaways:** Using time horizon, we observe that agent capability is more consistent with contractor time than experienced maintainer time (4.1, C.2), and that agents perform under their time horizon on more realistic tasks (H.2). By using human baselined time, we also quantify the degree to which agents can do impressive tasks but are unreliable on longer tasks (Figure 4). In the camera ready, we will also have an analysis of time horizons across 9 benchmarks in other domains like GPQA (scientific QA), AIME (competition math), RLBench (robotics), and OSWorld (graphical computer use). Models’ time horizons in other domains differ from their software time horizons, though we find similar rates of increase, and time horizon trends allow better forecasting of when automation in these other domains will be practical.

---

> > ### Comment · Reviewer_irnC · 2025-08-03
> >
> > Thanks for the helpful clarifications! I have updated my score. I encourage authors to proofread the next version carefully and try to mitigate potential misunderstandings.

---

### Official Review · Reviewer_m8QB · 2025-07-02

**Clarity:** 4
**Significance:** 3
**Originality:** 4
**Rating:** 4
**Confidence:** 4

**Summary:**

The paper "Measuring AI Ability to Complete Long Software Tasks" proposes a new metric called the 50% task completion time horizon, which quantifies how long a task takes a human to complete that an AI model can solve with 50% reliability. By benchmarking 13 frontier AI models from 2019 to 2025 on 170 software and research tasks, the authors find that this time horizon has been doubling approximately every seven months, indicating rapid progress in AI capabilities. The study uses datasets like HCAST, RE-Bench, and a new SWAA suite to evaluate tasks ranging from seconds to hours, comparing AI performance to skilled human baselines. Key drivers of improvement include better logical reasoning, tool use, and adaptability. However, AI agents still struggle with “messier” tasks that lack structure or clear feedback. Authors extrapolate the results to say that by 2028–2030, AI systems may autonomously complete software tasks that currently take humans a month.

**Questions:**

I think that answering the next questions could help to improve the paper:

How do you account for the potential exponential growth in task complexity over time, and could this affect the interpretation of the 50% completion time horizon metric?

What measures have you taken to ensure that the models evaluated were not exposed to benchmark data during training, and how do you mitigate potential contamination effects?

Could you analyze or report how the gap between 50% and 100% task completion evolves, especially for longer or more complex tasks?

Have you considered evaluating model performance on unseen or adversarial variants of the benchmarks to better assess generalization?

Can you provide evidence or case studies showing how well the 50% time horizon metric correlates with real-world software development performance?

**Ethical Concerns:**

["NO or VERY MINOR ethics concerns only"]

**Final Justification:**

I think that the paper should be accepted if the other reviewers agree.

**Limitations:**

I would recommend the authors to read the paper: Larger and more instructable language models become less reliable (https://www.nature.com/articles/s41586-024-07930-y) as maybe the models work better and better with complex problems but forget the simple and routine ones (much more useful) because the large models have become too focused on solving increasingly complex problems (leaving aside the comments of possible contamination).

**Paper Formatting Concerns:**

It is ok to me

**Quality:**

3

**Strengths And Weaknesses:**

Strengths
*.- Novel Metric for Long-Horizon Tasks: The introduction of the 50% task completion time horizon offers a fresh, intuitive way to measure AI progress on complex, real-world tasks, bridging the gap between toy benchmarks and practical capabilities.
*.- Empirical Breadth and Historical Scope: The study evaluates 13 models across six years (2019–2025) and 170 tasks, providing a rich longitudinal view of AI progress, especially in software engineering and research domains.
*.- Insightful Trend Analysis: The finding that AI capability (as measured by the 50% horizon) doubles every ~7 months is a compelling and quantifiable insight into the pace of AI advancement.

Weaknesses
*.- Neglect of Task Complexity Growth: The paper does not analyze whether the complexity of tasks increases over time. If task difficulty grows exponentially, the gap between 50% and 100% completion could also grow non-linearly, potentially biasing the metric and overstating progress.
*.- Potential Benchmark Contamination: There’s a risk that models have been exposed to benchmark data during training, especially given the popularity of certain datasets. This could inflate performance and obscure whether improvements stem from genuine generalization or memorization.
*.- Limited External Validity and Real-World Testing: While the benchmarks are diverse, the paper lacks a clear discussion of how well these results translate to real-world software development environments, where ambiguity, collaboration, and evolving requirements are common.

---

> ### Author Rebuttal · Authors · 2025-07-31
>
> Thanks for the detailed review! We’re glad that you find our core time horizon metric novel and intuitive, and that you appreciate the breadth of tasks and models used in our empirical evaluations. We respond to each of your questions below.
>
> 1. **Task complexity growth for longer task lengths:** We find that a logistic curve against log-task-length space is the best way to model an agent’s success, and that the slopes of such logistic curves are fairly similar for each agent. If longer tasks were exponentially more complex, we would see steeper and steeper slopes, with agents able to consistently handle (say) 1-hour tasks but almost never 2-hour tasks. Instead we find that the 50% time horizons are roughly 5x longer than 80% time horizons (Section 3.2.1), at least on our data. This can perhaps be reconciled by a model where exponentially longer tasks have exponentially more steps, but the failure rate of agents is also decreasing exponentially over time.
> 2. **Data contamination concerns:** Since RE-Bench was released in late 2024, HCAST in 2025, and we developed SWAA privately, the problems are either private or past the training cutoff date of all models we evaluated (o3’s cutoff is June 2024). As for potential future models, some of the tasks we evaluate on are now public. HCAST authors claim 35/91 of the solutions in a subset of the 97 we used are potentially memorizable. RE-Bench has a public repo, but due to the small number of tasks we think it’s unlikely that a lab would train on it specifically. So overall, we think data contamination is not a concern on the models in this paper but may become relevant in the future. In the camera-ready, we will also include time horizons on LiveCodeBench, which is continuously updated to prevent contamination.
> 3. **Gap between 50% and higher% task horizons:** It is difficult to measure time horizons at reliability close to 100%, since one would need hundreds of independent samples and <<1% label noise to distinguish between 98% and 99% accuracy. However, we observe that 80% horizons are improving at similar rates (Figure 17), and 80% horizons are roughly 5x shorter than 50% horizons on all models, with the curve perhaps slightly steeper on more capable models (Figure 4).
> 4. **Unseen/adversarial datasets and benchmarks:** As discussed in Q2, all of the datasets are likely unseen to models. About other benchmarks, data were designed to be realistic rather than adversarial. However, we have looked at other benchmarks like SWE-bench Verified (Section 4.1), as well as GPQA Diamond, LiveCodeBench, and AIME (will be in camera ready). As discussed in the paper, some models have much shorter time horizons on SWE-bench Verified; on GPQA, 1-2x longer; this paper’s tasks are roughly in the middle.
> 5. **Generalization to real-world software engineering performance:** We perform several external validity checks (Section 4): measuring performance on SWE-bench Verified (4.1), subsets of HCAST that are more realistic (F.2), and on our internal PRs, which we manually reviewed for code quality as well as correctness (C.2). It also seems to be empirically true that models with higher time horizons also perform better on other SWE benchmarks, and anecdotally are more helpful for software engineers. To more definitively answer this question, in the future we’re hoping to study the software engineering performance of models at a higher level of realism, using real-world in-situ software engineering tasks done by professional software engineers.
>
> **Limitations:**
> Thanks for referencing this work, we'll discuss and cite it! The referenced work is consistent with our qualitative analysis of failures (Table 2) where we observe that many failures of less capable agents are obvious, like repeated failed actions, whereas o1 makes mistakes that might be harder to supervise, like submitting answers without checking. We do observe that recent models like Claude 3.7 and o3 get near-perfect accuracy on tasks shorter than ~4 minutes (again see Figure 4), but we can’t rule out that performance of more capable models is lower on a different distribution of simple tasks.

---

### Comment · Area_Chair_YfAf · 2025-08-03
**Area chair's message**

Hi Reviewers,

As the rebuttals and other reviews are available, please read them, add your further comments if any, join the discussion, and adjust your review if applicable.
Thank you.

Regards
AC

---

### Author Response · Authors · 2025-08-07
**Summary of changes to paper**

Thanks again to the reviewers for engaging with our work!

We'd like to briefly summarize some of the changes we've made to the paper in response to feedback:
* We've added more discussion of the large differences between time horizons at different levels of reliability.
* We've clarified the tasks and agent setups we consider in the introduction. In particular, we've highlighted the fact that we study language model agents on agentic, multi-step software tasks.
* We've corrected several typos and formatting errors in the paper, including correcting the inconsistent number of models studied to reflect the true number (16 total, of which 12 are "frontier").
* We've added brief discussion of more recent related work that measures the autonomous capabilities of language model agents in relevant domains, including SWE-Lancer.

In addition, as mentioned in our reply to reviewer irnC, we've also performed an analysis of time horizons across 9 benchmarks in other domains like GPQA (scientific QA), AIME (competition math), RLBench (robotics), and OSWorld (graphical computer use). Models’ time horizons in other domains differ from their software time horizons, though we find similar rates of increase, and time horizon trends allow better forecasting of when automation in these other domains will be practical. We will incorporate this analysis in the camera-ready version of the paper, if accepted.

---

### Decision · Program_Chairs · 2025-09-17

**Decision:**

Accept (poster)

**Comment:**

The paper presents a perspective of 50% task completion time horizon and empirical analysis. The review identifies its value to the relevant problems and evaluation. The paper would be expected to clarify and enhance its final version by addressing the concerns about various aspects, in particular, the significance and originality of the work to the main track, the assumption, evaluation settings and robustness analysis of the metric, as well as other relevant factors and aspects such as complexity, benchmarking, and presentation issues.